# Training, Attitudes, and Practice (TAP) among healthcare professionals in the Nelson Mandela Bay municipality, South Africa: *A health promotion and disease prevention perspective*

**Herbert I. Melariri** [1,2] *, **Chester Kalinda** [1,3], **Moses J. Chimbari** [1,4]

1 College of Health Sciences, University of KwaZulu Natal, Howard College Campus, Durban, South Africa,
2 Eastern Cape Department of Health, Port Elizabeth Provincial Hospital, Gqeberha, South Africa,
3 University of Global Health Equity (UGHE), Bill and Joyce Cummings Institute of Global Health, Kigali, Rwanda, 4 Great Zimbabwe University, Masvingo, Zimbabwe

* melariri@gmail.com

**Data Availability Statement:** All relevant data are within the paper and its Supporting Information files.

## Abstract

### Background

Healthcare professionals (HCPs) play a pivotal role in ensuring access to quality healthcare of patients. However, their role in health promotion (HP) and disease prevention (DP) has not been fully explored. This study aimed at determining how training, attitude, and practice (TAP) of HCPs influence their practice of HP and DP.

### Methods

Data on TAP regarding HP and DP were collected from 495 HCPs from twenty-three hospitals in the study area using a standardized questionnaire. Bivariate, univariate, and multivariate analyses were conducted to describe how the TAP of HCPs influence their HP and DP practices. The analysis was further desegregated at the three levels of healthcare (primary, secondary and tertiary levels).

### Results

Most of the medical doctors 36.12% (n = 173), registered nurses 28.39% (n = 136), and allied health professionals (AHPs) 11.27% (n = 54) indicated the absence of coordinated HP training for staff in their facilities. Similarly, 32.93% (n = 193) of the HCPs, indicated having participated in HP or DP training. Among those that had participated in HP and DP training, benefits of training were positive behaviour, attributions, and emotional responses. When compared at the different levels of healthcare, enhanced staff satisfaction and continuing professional development for HP were statistically significant only at the tertiary healthcare level. Multivariate analysis showed a likelihood of reduced coordinated HP training for staff among medical doctors (Coef 0.15; 95% CI 0.07–0.32) and AHPs (Coef 0.24; 95% CI 0.10–0.59) compared to nurses. Furthermore, medical doctors (Coeff: 0.66; 95% CI: 0.46–0.94)

**Funding:** The author(s) received no specific funding for this work.

**Competing interests:** The authors have declared that no competing interests exist.

were less likely to agree that HCPs should model good health behavior to render HP services as compared to nurses.

## Conclusion

Training in HP and DP empowers HCPs with the requisite knowledge and attitude necessary for effective practice. Several HCPs at different levels of care had limited knowledge of HP and DP because of inadequate training. We recommend a strategy aimed at addressing the knowledge and attitudinal gaps of HCPs to ensure effective HP and DP services to patients.

## Introduction

Despite increasing awareness on HP and DP, their integration into healthcare practice remains a persistent challenge. Globally, mortality and morbidity from preventable and lifestyle-related diseases continue to rise. Every year, 41 million people die from non-communicable diseases (NCD) which is tantamount to 71% of all deaths worldwide [1] According to the World Health Organization (WHO), alcohol and tobacco abuse, physical inactivity and unwholesome diets are risk factors for NCD-related deaths [1]. Health screening, detection, and care form core response components to NCD. In 2019, there were 869 770 cases of measles and 207 500 deaths [2]. These deaths accounted for a 50% rise in four years [2]. Between 2017 and 2018, in the United States of America, the prevalence of adult obesity across both sexes was shown to be 42% [3]. Obesity poses a serious barrier to the prevention of chronic diseases globally [4]. Focus on the impact of HP and DP especially concerning physical activity and diet in alleviating the risk of obesity have gained more attention as no less than 2.8 million people die annually from either overweight or obesity [5]. In China, available data reveal that obesity is an independent and adjustable risk factor for diabetes mellitus [6]. A strong association has been demonstrated between obesity and hypertension [7, 8], coronary heart diseases [9, 10], atherosclerosis, and sudden cardiac death [10].

While the performance of routine clinical duties such as diagnosis, screening, patient care, and treatment are easily achieved by HCPs, the awareness [11], training, attitude, confidence, and consensus required to render HP and DP services are lacking. Several multifaceted impediments have been associated with poor or incoherence of HP and DP practice among HCPs. Evidence from the United Kingdom reveals that although HCPs are committed to delivering HP and DP services, they are limited by a lack of relevant training, inadequate resources, and time constraints [12]. In Ethiopia, limited training in HP negatively impacted the knowledge base of HCPs resulting in unsafe practices [13].

Categorized as an upper-middle-income nation, South Africa has high levels of unemployment and poverty [14]. In South Africa, morbidity and mortalities resulting from preventable and lifestyle modifiable diseases continue to soar. For three years (2015–2017), five out of ten leading causes of natural death in South Africa included tuberculosis, diabetes mellitus, human immunodeficiency (HIV) disease, hypertensive diseases, and ischemic heart diseases [15]; all of which can be prevented or controlled by adopting HP and DP practices. In an evaluation of risk factors that contribute to combined highest morbidity and disability, the Institute for Health Metrics and Evaluation showed that unsafe sex and malnutrition are the greatest contributors. Both risk factors contributed -41.6% and -33.5% change between 2009 and 2019 [16].

For HCPs to adequately meet the practice challenges posed by preventable and lifestyle modifiable diseases, there is need for institutionalized HP and DP training as well as the right attitudes. Available evidence shows that nurses [17] and physicians [18] who personally engage in HP and DP practices are more inclined to encourage their patients to act accordingly. Attitudes are of vital importance in health practice; they are an indicator of how HCPs perceive matters and reach a decision on what they consider appropriate.

### Rationale

Urgent intervention through implementation of HP and DP practices by HCPs is needed to halt the progression of morbidity and mortality due to many preventable diseases in South Africa. The feasibility of hospital-centered HP and DP services to patients is well documented [11, 18–21]. McMahon and Connolly recommended collaboration among disciplines to improve health through HP [22]. However, HCPs who engage in HP and DP practices do so in professional silos [17, 23] with no coordination.

The NMBM is the largest city in the Eastern Cape Province and a major economic player both for the province and South Africa as a whole [24]. Like the rest of South Africa, the municipality's healthcare system has all three levels of care comprising primary, secondary, and tertiary healthcare services. Despite the availability of inexpensive measures such as dietary counselling, physical activity, and handwashing for HP and DP, HCPs do not have structured training and knowledge to effectively implement the measures [13]. Furthermore, there is limited evidence on the HP and DP training and attitudes of HCPs concerning HP and DP especially in resource-constrained settings [25, 26]. Based on the literature review for this project, no baseline study has been conducted to both identify and address the gap in the municipality. It is therefore important to evaluate how HP and DP training and attitudes influence HCPs' practice to develop appropriate interventions. Thus, the purpose of this cross-sectional study was to determine the impact of HCPs' training and attitudes on their HP and DP practices and to compare the impact at various healthcare levels.

## Materials and methods

### Study setting

This study was conducted in the Nelson Mandela Bay Municipality (NMBM) of South Africa. For ease of primary healthcare delivery, the South African health system is decentralized into District Health Systems. While the national department of health is responsible for the policy mandate of the health system, health services provided at district levels are managed by the respective provincial health departments. The NMBM health district is divided into three sub-districts–A, B, and C with a total of 53 public primary healthcare facilities distributed across the three sub-districts.

The municipality has one secondary and three tertiary hospitals. The tertiary hospitals focus on clinical specialties separate from each other. The health system is two-pronged: public and private. While the private is mostly used by patients on medical aid and those that can pay for healthcare, the public system provides free services. The municipality has an estimated 3500 healthcare professionals [27] serving a population of about 1.24 million [24], with the majority depending on the free public health system.

### Study design and sample

This quantitative cross-sectional study was carried out among 520 HCPs that were randomly sampled from 23 hospitals in the NMBM. Only 501 HCPs completed the survey. Of the 501

respondents, six HCPs did not include their professions and were excluded from analysis. A total of 495 HCPs were included in the final analysis. An initial sample size of 384 was reached using a confidence level of 95% and an error margin of 5%. This sample size was further adjusted by 25% to 480 to account for non-responding HCPs. The hospitals included primary, secondary, and tertiary levels. The healthcare professionals and hospitals were randomly selected. Our sampling focused on HCPs that consult with patients in their hospitals. The study was carried out between January and March 2020. Gatekeepers of all sampled hospitals were formally notified in detail about the study. Their permissions were received before commencement. Our analysis was restricted to HCPs who consented to participate in the study.

## Study tool and data collection

A standardized questionnaire adapted from a previously validated study [23] was developed to determine the influence of HCPs' training and attitude on their practice of HP and DP. The questionnaire was further subjected to a pilot test. The questionnaire (provided as S1 File) was divided into three sections. Section one explored the demographics of the respondents (sex, registration status with the relevant professional board, profession, and level of hospital in which the respondent practices–primary, secondary, or tertiary). A primary-level hospital refers to a healthcare facility where ambulatory or first-contact personal health care services are provided [28] and in this context includes the Primary Healthcare Centres (PHC), community clinics, Comprehensive Health Centers (CHC), and Midwife Obstetrics units (MOU). A secondary-level hospital refers to a healthcare facility that is highly differentiated by function with five to ten clinical specialties [29]. This level of facility includes the Regional Hospitals, Provincial Hospitals, and General Hospitals. The inpatient bed capacity ranges from 200 to 800. Tertiary-level hospitals on the other hand are hospitals with highly specialized professionals and facilities such as cardiology, intensive care unit, and specialized imaging infrastructures among others [29]. The clinical services of the tertiary healthcare facilities are highly differentiated by function and have an inpatient bed capacity of 300 to 1500. This level of hospitals includes National Hospitals, Central Hospitals, and Academic, or teaching or University Hospitals.

The second section assessed institutional training structure in place as well as individual training efforts on HP and DP (e.g., availability of coordinated HP training for staff, continuing professional development for health promotion in the hospital, respondent's participation in HP training, benefits of HP training such as improved knowledge, improved skills, enhanced confidence, enhanced staff satisfaction, increased support to patients, positive staff behavior, attributions and emotional responses, and no added benefit). The third section assessed HCPs' attitudes towards HP and DP using eight questions related to their views of HP and DP within the healthcare system.

Responses of study participants were measured using the 5-point Likert scale (1 = strongly disagree to 5 = strongly agree) and categorical responses ('Yes', 'No' or 'I don't know'). Data collection was done using a pre-tested, paper-based questionnaire administered by postgraduate and nursing students who were trained on the data collection process, questionnaire contents, and ethical issues by the principal investigator (PI). Data was collected between January and March 2020.

## Ethical consideration

The Biomedical Research Ethics committee (BREC) of the University of KwaZulu Natal and the Eastern Cape Health Research committee approved the study protocol. Informed written

consent was received from each participant before commencement. Consenting participants were then enrolled in the study.

### Statistical analysis

Data summary tables were done using Microsoft Excel and statistical analysis was done using Stata. Univariate, bivariate, and multivariate analyses were used to compare HP and DP training, attitudes, and practices among the medical doctors, registered nurses, and allied health professionals (AHPs). Pearson Chi-square test was used to determine the association between categorical variables. Because data were collected from the healthcare facility levels (Primary, Secondary, and Tertiary Healthcare), our analysis controlled for healthcare level to determine the type of training, attitude, and practice that influenced HCPs stationed at different levels of healthcare. Multinomial logistic regression was used to determine the influence of training, attitude, and practice on HCPs.

## Results

### Socio-demographic characteristics

The response rate for this study was 95%, (n = 495). The majority of the HCPs (70.46%; n = 353) worked at tertiary hospitals, followed by HCPs at the primary hospitals (26.15%; n = 131). Healthcare professionals working at the secondary hospitals were the least (3.39%; n = 17). Participating healthcare professionals included medical doctors (38.79%; 192), registered nurses (47.27%; n = 234), and AHPs (13.94%; n = 69). The AHPs comprised social workers, speech therapists, occupational therapists, dieticians, and physiotherapists.

### Training in health promotion and disease prevention

Sixty-five (13.63%) medical doctors, 92 (19.29%) nurses, and 36 (7.55%) AHPs had been trained in HP. A significant association between the three groups of HCPs (Medical doctors, registered nurses, and allied health professionals) and the presence of coordinated HP and DP training for staff, previous participation in HP training, and availability of HP and DP related continuing professional development (CPD) programmes was noted (Table 1). Positive staff behavior, attributions, and emotional responses were found to be derived benefits of HP training. The associations between HP and HCPs are shown in Table 1.

### Comparison of HP and DP training and attitude of HCPs at the various levels of healthcare

Comparison of HP and DP training; and attitude of HCPs at primary, secondary and tertiary levels (Table 2) showed significant associations between HCPs at the tertiary healthcare level for the following variables: coordinated HP and DP training, previous participation in HP and DP training, and continuing HP and DP related professional development. No significant associations were found at both the primary and secondary levels. Associated training benefits with HCPs working at the tertiary healthcare levels included positive staff behavior, attributions, and emotional responses. Enhanced staff satisfaction was an additional training benefit associated with HCPs at the tertiary healthare levels. No HP or DP training benefit was significantly associated with HCPs at the primary and secondary healthcare levels.

Multivariate analysis showed that the risk of coordinated HP training for medical doctors (Coef 0.14; 95% CI 0.06–0.32) and AHPs (Coef 0.24; 95% CI 0.09–0.63) were both reduced compared to nurses. The presence of positive staff behavior, attributions, and emotional responses for AHPs (Coef 0.10.; CI 0.03–0.36) was equally lower than for nurses. No statistical

**Table 1. Bivariate analysis of training in health promotion and among the different healthcare professional groups.**

| Training in HP | Responses | Healthcare professionals (%) | | | p-value |
|---|---|---|---|---|---|
| | | Medical doctors | Registered Nurses | AHPs | |
| Is there a coordinated HP training for staff? | No | 36.12% (n = 173) | 28.39% (n = 136) | 11.27% (n = 54) | **0.000** |
| | Yes | 3.55% (n = 17) | 17.54% (n = 84) | 3.13% (n = 15) | |
| Have you ever participated in any HP training? | No | 25.99% (n = 124) | 27.04% (n = 129) | 6.50% (n = 31) | **0.019** |
| | Yes | 13.63% (n = 65) | 19.29% (n = 92) | 7.55% (n = 36) | |
| Benefits of training | | | | | |
| Improved knowledge | No | 27.55% (n = 54) | 44.90% (n = 88) | 15.82% (n = 31) | 0.733 |
| | Yes | 5.49% (n = 9) | 5.10% (n = 10) | 2.04% (n = 4) | |
| Improved skills | No | 8.67% (n = 17) | 8.67% (n = 17) | 6.12% (n = 12) | 0.093 |
| | Yes | 23.47% (n = 46) | 41.33% (n = 81) | 11.73% (n = 23) | |
| Enhanced confidence | No | 10.77% (n = 21) | 14.36% (n = 28) | 8.21% (n = 16) | 0.193 |
| | Yes | 21.54% (n = 42) | 35.38% (n = 69) | 9.74% (n = 19) | |
| Positive staff behavior, attributions, and emotional responses | No | 16.84% (n = 33) | 20.92% (n = 41) | 14.29% (n = 28) | **0.001** |
| | Yes | 15.01% (n = 30) | 29.08% (n = 57) | 3.57% (n = 7) | |
| Enhanced staff satisfaction | No | 21.94% (n = 43) | 28.57% (n = 56) | 13.78% (n = 27) | 0.077 |
| | Yes | 10.20% (n = 20) | 21.43% (n = 42) | 4.08% (n = 8) | |
| Increased support to patients | No | 19.39% (n = 38) | 34.69% (n = 68) | 9.69% (n = 19) | 0.220 |
| | Yes | 12.76% (n = 25) | 10.20% (n = 20) | 8.16% (n = 16) | |
| Enhanced staff retention | No | 25.00% (n = 49) | 36.22% (n = 71) | 14.29% (n = 28) | 0.591 |
| | Yes | 7.14% (n = 14) | 13.78% (n = 27) | 3.57% (n = 7) | |
| No added benefit | No | 31.63% (n = 62) | 48.47% (n = 95) | 16.33% (n = 32) | 0.189 |
| | Yes | 0.51% (n = 1) | 1.53% (n = 3) | 1.53% (n = 3) | |
| Is there a continuing professional development for health promotion in your facility | No | 13.29% (n = 63) | 11.39% (n = 54) | 3.38% (n = 16) | **0.000** |
| | Yes | 5.27% (n = 25) | 23.84% (n = 113) | 3.38% (n = 16) | |
| | I don't know | 21.10% (n = 100) | 11.18% (n = 53) | 7.17% (n = 34) | |

difference was observed for staff participation in training, improved knowledge, improved skills, enhanced confidence, and staff satisfaction.

In the final model (Table 3), there was a likelihood of reduced coordinated HP training for staff among medical doctors (Coef 0.15; 95% CI 0.07–0.32) and AHPs (Coef 0.24; 95% CI 0.10–0.59) compared to nurses. Furthermore, the likelihood of positive staff behavior, attributions, and emotional responses for AHPs (Coef 0.17; 95% CI 0.07–0.45) was lower compared to nurses.

## Attitudes towards health promotion and disease prevention

Analysis of HCPs' attitudes towards HP and DP showed that most of the HC professionals (98.72%; n = 462) expressed that it was important to participate in HP and DP. In addition, 21.02% (n = 103) of the medical doctors and 21.02% (n = 103) of the registered nurses, disagreed that patients did not want health education (HE) from HCPs while among the AHPs, 6.73% (n = 33) disagreed that patients did not want HE from HCPs. Table 4 shows the association between the health care professional groups and attitudes towards health promotion variables.

Following a bivariate analysis comparing attitudes towards HP and the various HCP groups, five variables were found to be associated with HCPs practice: HCPs should model good health behavior to give HP advice; HCPs should be encouraged to engage in HP as part of government policy and healthcare services; patients who deliberately engage in an unhealthy lifestyle will not benefit from health promotion; health education, advise and counseling from HCPs could positively enhance patients' health; and I do not have time to implement HP.

**Table 2. Bivariate analysis of HP and DP training and attitude of HCPs at different levels of healthcare.**

| HP Training | Responses | Primary Health Care level | | | | Secondary Health Care level | | | | Tertiary Health Care level | | | |
|---|---|---|---|---|---|---|---|---|---|---|---|---|---|
| | | Medical doctors | Nurses | AHPs | p-value | Medical doctors | Nurses | AHPs | p-value | Medical doctors | Nurses | AHPs | P-value |
| Is there a coordinated HP training for staff? | No | 3.20% (n = 4) | 52.00% (n = 65) | 4.00% (n = 5) | 0.976 | 37.50% (n = 6) | 6.25% (n = 1) | 31.25% (n = 5) | 0.90 | 48.22% (n = 163) | 20.71% (n = 70) | 4.14% (n = 14) | **0.000** |
| | Yes | 2.40% (n = 3) | 36.00% (n = 45) | 2.40% (n = 3) | | 0.00% (n = 0) | 12.50% (n = 2) | 12.50% (n = 2) | | 4.14% (n = 14) | 10.95% (n = 37) | 2.96% (n = 10) | |
| Have you ever participated in any HP training? | No | 3.20% (n = 4) | 44.80% (n = 56) | 4.80% (n = 6) | 0.408 | 31.25% (n = 5) | 12.50% (n = 2) | 31.25% (n = 5) | 0.827 | 34.23% (n = 115) | 21.13% (n = 71) | 5.95% (n = 20) | **0.001** |
| | Yes | 2.40% (n = 3) | 43.20% (n = 54) | 1.60% (n = 2) | | 6.25% (n = 1) | 6.25% (n = 1) | 12.50% (n = 2) | | 18.15% (n = 61) | 11.01% (n = 37) | 9.52% (n = 32) | |
| Benefits of training | | | | | | | | | | | | | |
| Improved knowledge | No | 4.76% (n = 3) | 80.95% (n = 51) | 1.59% (n = 1) | 0.700 | 25.00% (n = 1) | 25.00% (n = 1) | 50.00% (n = 2) | | 38.76% (n = 50) | 27.91% (n = 36) | 21.71% (n = 28) | 0.645 |
| | Yes | 1.59% (n = 1) | 11.11% (n = 7) | 0.00% (n = 0) | | 0.00% (n = 0) | 0.00% (n = 0) | 0.00% (n = 0) | | 6.20% (n = 8) | 2.33% (n = 3) | 3.10% (n = 4) | |
| Improved skills | No | 0.00% (n = 0) | 17.46% (n = 11) | 1.59% (n = 1) | 0.075 | 0.00% (n = 0) | 25.00% (n = 1) | 25.00% (n = 1) | 0.368 | 13.18% (n = 17) | 3.88% (n = 5) | 7.75% (n = 10) | 0.114 |
| | Yes | 6.35% (n = 4) | 74.60% (n = 47) | 0.00% (n = 0) | | 25.00% (n = 1) | 0.00% (n = 0) | 25.00% (n = 1) | | 31.78% (n = 41) | 26.36% (n = 34) | 17.05% (n = 22) | |
| Enhanced confidence | No | 1.59% (n = 1) | 25.40% (n = 16) | 1.59% (n = 1) | 0.279 | 0.00% (n = 0) | 25.00% (n = 1) | 50.00% (n = 2) | 0.135 | 15.63% (n = 20) | 8.59% (n = 11) | 10.16% (n = 13) | 0.591 |
| | Yes | 4.76% (n = 3) | 66.67% (n = 42) | 0.00% (n = 0) | | 25.00% (n = 1) | 0.00% (n = 0) | 0.00% (n = 0) | | 29.69% (n = 38) | 21.09% (n = 27) | 14.84% (n = 19) | |
| Positive staff behavior, attributions and emotional responses | No | 4.76% (n = 3) | 38.10% (n = 24) | 1.59% (n = 1) | 0.225 | 0.00% (n = 0) | 0.00% (n = 0) | 50.00% (n = 2) | 0.135 | 23.26% (n = 30) | 13.17% (n = 17) | 19.38% (n = 25) | **0.010** |
| | Yes | 1.59% (n = 1) | 53.97% (n = 34) | 0.00% (n = 0) | | 25.00% (n = 1) | 25.00% (n = 1) | 0.00% (n = 0) | | 21.71% (n = 28) | 17.05% (n = 22) | 5.43% (n = 7) | |
| Enhanced staff satisfaction | No | 4.76% (n = 3) | 57.14% (n = 36) | 1.59% (n = 1) | 0.652 | 0.00% (n = 0) | 25.00% (n = 1) | 50.00% (n = 2) | 0.135 | 31.01% (n = 40) | 14.73% (n = 19) | 18.60% (n = 24) | **0.043** |
| | Yes | 1.59% (n = 1) | 34.92% (n = 22) | 0.00% (n = 0) | | 25.00% (n = 1) | 0.00% (n = 0) | 0.00% (n = 0) | | 13.95% (n = 18) | 15.50% (n = 20) | 6.20% (n = 8) | |
| increased support to patients | No | 1.59% (n = 1) | 61.90% (n = 39) | 1.59% (n = 1) | 0.175 | 25.00% (n = 1) | 25.00% (n = 1) | 25.00% (n = 1) | 0.513 | 27.91% (n = 36) | 21.70% (n = 28) | 13.18% (n = 17) | 0.266 |
| | Yes | 4.76% (n = 3) | 30.16% (n = 19) | 0.00% (n = 0) | | 0.00% (n = 0) | 0.00% (n = 0) | 25.00% (n = 1) | | 17.05% (n = 22) | 8.53% (n = 11) | 11.63% (n = 15) | |
| Enhanced staff retention | No | 4.76% (n = 3) | 65.08% (n = 41) | 1.59% (n = 1) | 0.802 | 0.00% (n = 0) | 25.00% (n = 1) | 50.00% (n = 2) | 0.135 | 35.66% (n = 46) | 22.48% (n = 29) | 19.38% (n = 25) | 0.845 |
| | Yes | 1.59% (n = 1) | 26.98% (n = 17) | 0.00% (n = 0) | | 25.00% (n = 1) | 0.00% (n = 0) | 0.00% (n = 0) | | 9.30% (n = 12) | 7.75% (n = 10) | 5.43% (n = 7) | |
| No added benefit | No | 6.35% (n = 4) | 88.89% (n = 56) | 1.59% (n = 1) | 0.915 | 25.00% (n = 1) | 25.00% (n = 1) | 50.00% (n = 2) | | 44.19% (n = 57) | 29.46% (n = 38) | 22.48% (n = 29) | 0.174 |
| | Yes | 0.00% (n = 0) | 3.17% (n = 2) | 0.00% (n = 0) | | 0.00% (n = 0) | 0.00% (n = 0) | 0.00% (n = 0) | | 0.78% (n = 1) | 0.78% (n = 1) | 2.33% (n = 3) | |
| Is there a continuing professional development for health promotion in your facility | No | 2.42% (n = 3) | 27.42% (n = 34) | 2.42% (n = 3) | 0.738 | 18.75% (n = 3) | (n = 1) | 12.50% (n = 2) | 0.247 | 17.07% (n = 57) | 5.69% (n = 19) | 3.29% (n = 11) | **0.000** |
| | Yes | 1.61% (n = 2) | 39.52% (n = 49) | 1.61% (n = 2) | | 0.00% (n = 0) | 12.50% (n = 2) | 18.75% (n = 3) | | 6.89% (n = 23) | 18.56% (n = 62) | 3.29% (n = 11) | |
| | I don't know | 1.61% (n = 2) | 20.97% (n = 26) | 2.42% (n = 3) | | 18.75% (n = 3) | 0.00% (n = 0) | 12.50% (n = 2) | | 28.44% (n = 95) | 8.08% (n = 27) | 8.86% (n = 29) | |

Results from the multinomial logistic regression analysis (Table 5) showed that medical doctors (Coeff: 0.66; 95% CI: **0.46–0.94)** were less likely to agree that HCPs should model good health behavior to give HP advice compared to nurses. Furthermore, medical doctors (Coef 0.56; 95% CI 0.40–0.81) were less likely to view health promotion as a waste of time. In

**Table 3. Multivariate analysis between HCP and HP training.**

| HP training | Healthcare Professions | Coef (unadjusted) | P value | 95% CI | Coef (adjusted) | P value | 95% CI |
|---|---|---|---|---|---|---|---|
| Is there a coordinated HP training for staff? | MD | **0.14** | **0.000** | **0.61–0.32** | **0.15** | **0.000** | **0.07–0.32** |
| | AHPs | **0.25** | **0.006** | **0.09–067** | **0.24** | **0.002** | **0.10–0.59** |
| Have you ever participated in any HP training? | Medical doctors | 2.48 | 0.222 | 0.58–10.66 | 1.94 | 0.351 | 0.48–7.77 |
| | AHPs | 2.28 | 0.391 | 0.35–15.04 | 2.10 | 0.409 | 0.36–12.19 |
| Improved knowledge | Medical doctors | 0.52 | 0.303 | 0.15–1.82 | | | |
| | AHPs | 1.26 | 0.783 | 0.25–6.02 | | | |
| Improved skills | Medical doctors | 0.75 | 0.567 | 0.27–2.04 | | | |
| | AHPs | **0.52** | **0.268** | 0.17–1.65 | | | |
| Enhanced confidence | Medical doctors | 1.56 | **0.358** | **0.60–4.03** | | | |
| | AHPs | 1.37 | **0.568** | **0.46–4.08** | | | |
| Positive staff behavior, attributions, and emotional responses | Medical doctors | 0.68 | 0.417 | 0.27–1.73 | 0.63 | 0.197 | 0.31–1.27 |
| | AHPs | **0.11** | **0.002** | **0.03–0.45** | **0.17** | **0.000** | **0.07–0.45** |
| Enhanced staff satisfaction | Medical doctors | 0.67 | 0.44 | 0.24–1.85 | | | |
| | AHPs | 0.62 | 0.465 | 0.18–2.21 | | | |
| increased support to patients | Medical doctors | 0.69 | 0.407 | 0.29–1.64 | | | |
| | AHPs | 0.93 | 0.884 | 0.34–2.53 | | | |
| Enhanced staff retention | Medical doctors | 2.01 | 0.245 | 0.62–6.56 | 1.30 | 0.598 | 0.49–3.44 |
| | AHPs | 3.67 | 0.098 | 0.79–17.18 | 2.68 | 0.155 | 0.69–10.44 |
| No added benefit | Medical doctors | 0.37 | 0.472 | 0.24–5.61 | 0.49 | 0.601 | 0.04–6.94 |
| | AHPs | 4.21 | 0.230 | 0.44–40.57 | 4.30 | 0.197 | 0.47–39.56 |
| Is there a continuing professional development for health promotion in your facility | Medical doctors | **1.76** | **0.032** | **1.05–2.95** | 1.60 | 0.063 | 0.98–2.64 |
| | AHPs | 1.38 | 0.323 | 0.73–2.59 | 1.35 | 0.338 | 0.73–2.50 |

addition, medical doctors (Coef 0.79; 95% CI 0.66–0.96) were less likely to view patients who deliberately engaged in an unhealthy lifestyle as benefitting from health promotion.

## Discussion

Using data obtained from several hospitals in the Nelson Mandela Bay Municipality, we identified training and attitudinal factors among various healthcare professional groups (medical

**Table 4.  Bivariate analysis of attitudes towards health promotion and each healthcare professional group.**

| Attitudes towards HP | Responses | Healthcare workers by professions (%) | | | p-value |
|---|---|---|---|---|---|
| | | **Medical doctors** | **Nurses** | **AHPs** | |
| HCW should model good health behavior in order to give HP advice | Strongly disagree | 0.61% (n = 3) | 0.61% (n = 3) | 0.20% (n = 1) | **0.001** |
| | Disagree | 1.43% (n = 7) | 0.00% (n = 0) | 0.20% (n = 1) | |
| | Neutral | 4.50% (n = 22) | 2.21% (n = 10) | 1.43% (n = 7) | |
| | Agree | 19.43% (n = 95) | 19.43% (n = 95) | 6.95% (n = 34) | |
| | Strongly agree | 13.29% (n = 65) | 24.74% (n = 121) | 5.11% (n = 25) | |
| HCW should be encouraged to engage in HP as part of government policy and healthcare services | Strongly disagree | 0.61% (n = 3) | 0.61% (n = 3) | 0.21% (n = 1) | **0.008** |
| | Disagree | 0.82% (n = 4) | 0.41% (n = 2) | 0.21% (n = 1) | |
| | Neutral | 5.93% (n = 29) | 2.25% (n = 11) | 14.31% (n = 7) | |
| | Agree | 18.20% (n = 89) | 19.83% (n = 97) | 7.16% (n = 35) | |
| | Strongly agree | 13.70% (n = 67) | 23.72% (n = 116) | 4.90% (n = 24) | |
| Health promotion is a waste of time | Strongly disagree | 6.64% (n = 32) | 31.74% (n = 153) | 11.20% (n = 54) | 0.358 |
| | Disagree | 10.17% (n = 49) | 11.20% (n = 54) | 2.28% (n = 11) | |
| | Neutral | 1.24% (n = 6) | 1.25% (n = 6) | 0.21% (n = 1) | |
| | Agree | 0.42% (n = 2) | 1.45% (n = 7) | 0.42% (n = 2) | |
| | Strongly agree | 0.00% (n = 0) | 0.83% (n = 4) | 0.21% (n = 1) | |
| Patients who deliberately engage in an unhealthy lifestyle will not benefit from health promotion | Strongly disagree | 12.32% (n = 60) | 13.96% (n = 68) | 3.90% (n = 19) | **0.002** |
| | Disagree | 16.02% (n = 78) | 13.37% (n = 65) | 5.96% (n = 29) | |
| | Neutral | 4.92% (n = 24) | 3.49% (n = 17) | 1.23% (n = 6) | |
| | Agree | 4.52% (n = 22) | 10.27% (n = 50) | 1.43% (n = 7) | |
| | Strongly agree | 1.62% (n = 8) | 5.54% (n = 27) | 1,43% (n = 7) | |
| Health education, advise and counseling from HCW could positively enhance patients' health | Strongly disagree | 1.03% (n = 5) | 1.23% (n = 6) | 0.21% (n = 1) | **0.042** |
| | Disagree | 0.82% (n = 4) | 1.22% (n = 6) | 0.21% (n = 1) | |
| | Neutral | 1.43% (n = 7) | 1.43% (n = 7) | 0.41% (n = 2) | |
| | Agree | (n = 102) | (n = 82) | (n = 25) | |
| | Strongly agree | 15.16% (n = 74) | 25.82% (n = 126) | 8.20% (n = 40) | |
| I do not have time to implement health promotion | Strongly disagree | 4.49% (n = 22) | 17.34% (n = 85) | 3.67% (n = 18) | **0.000** |
| | Disagree | 13.57% (n = 67) | 20.41% (n = 100) | 5.31% (n = 26) | |
| | Neutral | 9.59% (n = 47) | 4.08% (n = 20) | 1.84% (n = 9) | |
| | Agree | 8.98% (n = 44) | 3.06% (n = 15) | 2.45% (n = 12) | |
| | Strongly agree | 2.24% (n = 11) | 2.04% (n = 10) | 0.82% (n = 4) | |
| Patients do not want health education from HCW | Strongly disagree | 8.78% (n = 43) | 13.27% (n = 65) | 1.84% (n = 9) | 0.129 |
| | Disagree | 21.02% (n = 103) | 21.02% (n = 103) | 6.73% (n = 33) | |
| | Neutral | 5.51% (n = 27) | 7.76% (n = 38) | 3.47% (n = 17) | |
| | Agree | 2.04% (n = 10) | 3.47% (n = 17) | 1.43% (n = 7) | |
| | Strongly agree | 1.43% (n = 7) | 1.63% (n = 8) | 0.61% (n = 3) | |
| Do you think HCW should participate in HP? | No | 0.21% (n = 1) | 1.07% (n = 5) | 0.00% (n = 0) | 0.181 |
| | yes | 38.89% (n = 182) | 45.51% (n = 213) | 14.32% (n = 67) | |

**Table 5. Multivariate analysis of attitudes towards health promotion.**

| Attitudes towards Hp and DP | Healthcare Professions | Coef (unadjusted) | P value | 95% CI | Coef (adjusted) | P value | 95% CI |
|---|---|---|---|---|---|---|---|
| HCPs should model good health behavior in order to give HP advice | MD | **0. 68** | **0.043** | **0.46–0.99** | **0.66** | **0.020** | **0.46–0.94** |
| | AHPs | 0.69 | 0.122 | 0.43–1.11 | 0.69 | 0.117 | 0.44–1.09 |
| I do not have time to implement health promotion | Medical doctors | **2.39** | **0.000** | **1.88–3.03** | **2.27** | **0.000** | **1.81–2.84** |
| | AHPs | **1.74** | **0.000** | **1.29–2.33** | **1.62** | **0.001** | **1.22–2.17** |
| Patients do not want health education from HCPs | Medical doctors | 0.86 | 0.280 | 0.66–1.13 | 0.93 | 0.592 | 0.73–1.20 |
| | AHPs | **1.43** | **0.023** | **1.05–1.95** | **1.40** | **0.026** | **1.04–1.88** |
| Do you think HCPs should participate in HP? | Medical doctors | 9.43 | 0.053 | 0.97–91.56 | | | |
| | AHPs | 0.13 | 0.985 | | | | |
| HCPs should be encouraged to engage in HP as part of government policy and healthcare services | Medical doctors | 0.66 | 0.030 | 0.46–0.96 | 0.69 | 0.038 | 0.49–0.98 |
| | AHPs | **0.63** | **0.051** | 0.39–1.00 | .74 | 0.180 | 0.48–1.15 |
| Health promotion is a waste of time | Medical doctors | **0.55** | **0.002** | **0.38–0.80** | **0.56** | **0.002** | **0.40–0.81** |
| | AHPs | **0.56** | **0.023** | **0.34–0.92** | **0.51** | **0.007** | **0.31–0.83** |
| Patients who deliberately engage in an unhealthy lifestyle will not benefit from health promotion | Medical doctors | **0.79** | **0.019** | **0.65–0.96** | **0.79** | **0.015** | **0.66–0.96** |
| | AHPs | 0.88 | 0.305 | 0.69–1.12 | 0.87 | 0.293 | 0.69–1.12 |
| Health education, advise and counselling from HCW could positively enhance patients' health | Medical doctors | 0.87 | 0.354 | 0.64–1.17 | | | |
| | AHPs | 1.42 | 0.141 | 0.89 2.27 | | | |

Registered Nurses as base outcome.

doctors, allied health professionals, and nurses) that influence their practice of health promotion and disease prevention at the different levels of healthcare services. For each professional group, there were different determinants relating to their practice of HP and DP that require different mitigatory measures. Our results show a positive correlation between HCPs' attitudes towards HP and their practice. However, training infrastructures at the hospitals and HCPs participation in HP and DP training were low.

## Training

Seventy five point seven eight percent (75.78%; n = 363) of participants indicated that there was no coordinated HP training for healthcare professionals at their hospitals. This was consistent with the findings of Cancedda et al 2015 [30]. This may signify that the relevance of adequate HP training for healthcare professionals is being ignored. Such an omission may seriously impact the quality of HP services or outrightly neglect it and may also result in the demoralization of professionals and the ability of the hospitals to retain good health professionals. Possible reasons for poor coordinated training may include difficulties in initiating such programs, wide-ranging negotiations with leadership, lack of resources and knowledge or

processes necessary for program initiation [30]. Health promotion and DP objectives such as promotion of a healthy lifestyle, prompt detection, and monitoring of disease can be achieved through coordinated training of healthcare workers. Such training should aim at equipping HCPs with the requisite skills to empower the population on living healthy [31].

Most of our study respondents had never participated in any HP or DP training. Despite the overall limited training of all health professionals, the AHPs reported greater participation in HP and DP training compared to medical doctors and nurses. Multivariate analysis further revealed that the AHPs were 2.10 times more involved in HP and DP training than the nurses. This compares well to previous studies which showed above 70% positive responses to training and knowledge in health promotion among physiotherapists [32, 33]. The low participation of medical doctors and registered nurses in HP and DP training in the present study may reflect their limited participation in HP and DP practice in general. A precise comparison of the training results from this study to others may prove challenging as a result of variation in the sample population (most studies explored KAP in a specific health care profession) [23, 34, 35]. The positive staff behavior, attributions, and emotional responses among AHPs could be explained by their greater participation in relevant training [36].

In addition to coordinated HP training, previous participation in HP training, and CPD, the study revealed a further statistically significant association between two training benefits (enhanced staff satisfaction and positive behavior) and HCPs at the tertiary hospitals only. This finding could not be compared to any other study as there were no other studies comparing TAP of HP and DP at the various levels of healthcare facilities found during the literature search for this study.

## Attitudes

Consistent with existing literature [37–39], our analysis revealed an overall acceptance that HCPs should serve as role models of good health behaviors before offering such services to patients. Health promotion and disease prevention strategies like physical fitness and good nutrition positively influenced HCPs' health and wellbeing, as well as their HP and DP practice [40]. It follows, therefore, that efforts towards increasing the knowledge of HCPs' lifestyle and detection of possible limitations to their lifestyle changes such as requisite stimuli, may be essential elements in the pursuit of HP and DP of the general population. After controlling for the attitudinal variables at the different levels of healthcare, our findings indicated that HCPs serving as role models for good health behavior influence the practice of HCPs at the primary healthcare levels compared to those at the secondary and tertiary healthcare levels. This may be related to the traditional intentions for Primary Healthcare Centers (PHC) that focus on population needs along the continuum from HP and DP [28]. Past studies [41, 42] in South Africa reported overstretching of public healthcare facilities at all levels leading to a spillover of patients meant for primary levels of care to both secondary and tertiary levels. The impact of this spillover threatens the quality of healthcare at all levels, with HCPs at secondary and tertiary levels not adequately prepared to meet the HP and DP needs of patients who are still part of the general population. Thus, interventions targeting HP and DP for HCPs at all levels of care are a necessary investment for healthcare systems.

In our study, only 38.03% (n = 132)of HCPs (p<0.001) at the tertiary hospitals agreed to having adequate time for HP and DP. Similarly, in a Dutch study [43], general practitioners, and practice nurses reported lack of time as a reason for not partaking in health promotion practices. The lack of time may be attributed to practice-related issues [43] and a high volume of patients. Our study further showed that among medical doctors, patients that continually live unhealthy lifestyle constituted a barrier to HP and DP. This finding corroborates with the study of Geense et al [43] which showed that some patients often lie about their true lifestyle.

Our study further identified that Patients not wanting health education from healthcare professionals emerged as an influencing variable among AHPs for HP and DP practice. HCPs are continually challenged to ensure that relevant information is communicated to patients to make an informed decision. Failure to receive this information may in part be attributed to patients' low level of health literacy [43]. Poor health literacy level negatively affects patient outcomes, and this unfortunately is often unrecognized. In many instances, patients are ashamed to notify the HCP that they either cannot read or comprehend information handed to them. The study of Veenker et al 2016 [44] recommended the use of scaffolding by a health practitioner in building an autonomy spiral. Such practice helps both players (HCP and patient) in building a long-term dynamic partnership of learning [44]. Other possible explanations for patients rejecting HE from HCPs include language barriers and cultural reasons. The realization of the consequences of patients not wanting HE from HCPs opens a window of opportunity to better understand the reasons for untoward outcomes and develop appropriate interventions to address them.

Patients who deliberately engage in an unhealthy lifestyle will not benefit from health promotion was one independent attitude predictor among medical doctors (adjusted Coef 0.79; 95% CI 0.66–0.96). This relationship may create an opportunity to developing patients' health literacy and participation through health promotion. Medical doctors will need to invest time and resources to actualize behavioral change that results in better population health outcomes. The medical doctor-patient relationship necessitates a dual obligation in which the medical doctor advises the patient on the process to reach his/her health goals, and the patient must comply with the information provided [45] for the best outcome.

## Strengths and limitations

Based on the literature review for this project, no previously published study has investigated the influence of training and attitudes on HCPs' practice of HP and DP in the NMBM. Also, the authors are not aware of any study that has compared the TAP of HCPs practice of HP and DP at the three levels of healthcare (primary, secondary, and tertiary levels). By fitting with the multinomial logistic regression model, we were able to reduce standard errors when compared to binary logistic regression used in previous studies.

Our study limitations include the possibility of information bias emanating from HCPs' self-reporting of information. The risk of recall bias or poor memory resulting in over or under-reporting of available training infrastructure abound. The confinement of the study to only one South African Province as well as the non-inclusion of HCPs in the private sector limits generalization of the study results. Some provinces have more fiscal capacity than others, and such provinces would most likely make more investment in healthcare delivery than the resource-depleted ones. Similarly, a wide gap exists between the private and public healthcare systems in South Africa. The two sectors are divided along socioeconomic lines. Unlike the public health sector that is government-funded, the private health sector is funded by individuals buying into expensive medical aid schemes, making the sector centers of quality care, better infrastructure, and relatively sufficient resources. Hence the situation between private and public hospitals is different, making it difficult to directly apply findings of our study to the private sector. The study was cross-sectional and therefore did not allow causal inferences.

## Recommendations

Our recommendations include expansion of training curriculum of tertiary institutions to cover a significant amount of HP and DP courses. Also, emphasis should be placed on in-service training of healthcare professionals on HP and DP continuously. A robust and

enthusiastic engagement between healthcare professionals and patients is needed for the realization of effective HP and DP interventions. Finally, HCPs should be encouraged to model good health and lifestyle practices before advising patients.

## Conclusion

This study provides an understanding of the impact of training and attitudes to healthcare professionals' practice of health promotion and disease prevention in the Nelson Mandela Bay Municipality of South Africa. Our findings show that HCPs in the NMBM have a positive attitude towards HP and DP but their training in these fields limits them to effectively practice. More nurses than medical doctors and allied health professionals reportedly had positive behavior, attribution, and emotional responses and this may be associated with the HP and DP training this set of nurses may have received.

The major impediments to training reported by the HCPs included a lack of training infrastructure in healthcare facilities. When controlled at the facility levels, the tertiary health facilities were the neediest of all three, with medical doctors and nurses at this level needing more training than the AHPs in HP and DP. The findings suggest the need for establishing HP and DP training programs in healthcare facilities, especially in those where HCPs reported limited training to fill the knowledge and attitudinal gaps. This study further suggests that training should be tailored to meet each healthcare professionals' need as these vary with each professional group.

The content and knowledge of this study may serve as a guide to health systems' managers and political leaders in the planning and implementation of relevant programs aimed at improving population health. Future observational studies to validate the self-reported data retrieved from HCPs should be considered.

## Supporting information

**S1 File. Questionnaire.**
(DOCX)

**S2 File. Dataset.**
(XLSX)

## Acknowledgments

We would like to thank all healthcare professionals who agreed to participate in this study. In addition, we are grateful to the Eastern Cape Department of Health, the Nelson Mandela Bay Municipality District Health, all tertiary, secondary, and primary healthcare facilities in the NMBM for granting us access to their institutions.

## Author Contributions

**Conceptualization:** Herbert I. Melariri.

**Data curation:** Herbert I. Melariri, Moses J. Chimbari.

**Formal analysis:** Herbert I. Melariri, Chester Kalinda, Moses J. Chimbari.

**Investigation:** Herbert I. Melariri.

**Methodology:** Herbert I. Melariri, Chester Kalinda, Moses J. Chimbari.

**Project administration:** Herbert I. Melariri, Moses J. Chimbari.

**Supervision:** Moses J. Chimbari.

**Writing – original draft:** Herbert I. Melariri.

**Writing – review & editing:** Herbert I. Melariri, Chester Kalinda, Moses J. Chimbari.

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
