## [Decision Letter · Decision Letter 0]

5 Oct 2021

PONE-D-21-07503Training, Attitudes, and Practice (TAP) among healthcare professionals in the Nelson Mandela Bay municipality, South Africa: A Health Promotion and Disease prevention perspectivePLOS ONE

Dear Dr. Melariri,

Thank you for submitting your manuscript to PLOS ONE. After careful consideration, we feel that it has merit but does not fully meet PLOS ONE’s publication criteria as it currently stands. Therefore, we invite you to submit a revised version of the manuscript that addresses the points raised during the review process.

The manuscript has been evaluated by two reviewers, and their comments are available below.

The reviewers have raised some concerns, including copyediting/language to improve readability to international audiences, rationale for experimental design, reporting of the study methodology, addressing experimental/survey bias, and data presentation.

Could you please revise the manuscript to carefully address the concerns raised?

We look forward to receiving your revised manuscript.

Kind regards,

Sebastian Shepherd

Associate Editor

PLOS ONE

Journal Requirements:

3. Please include additional information regarding the survey or questionnaire used in the study and ensure that you have provided sufficient details that others could replicate the analyses. For instance, if you developed a questionnaire as part of this study and it is not under a copyright more restrictive than CC-BY, please include a copy, in both the original language and English, as Supporting Information. Moreover, please include more details on how the questionnaire was pre-tested, and whether it was validated. 

Reviewers' comments:

Reviewer's Responses to Questions

**Comments to the Author**

1. Is the manuscript technically sound, and do the data support the conclusions?

Reviewer #1: Yes

Reviewer #2: Yes

2. Has the statistical analysis been performed appropriately and rigorously? 

Reviewer #1: Yes

Reviewer #2: Yes

3. Have the authors made all data underlying the findings in their manuscript fully available?

Reviewer #1: Yes

Reviewer #2: Yes

4. Is the manuscript presented in an intelligible fashion and written in standard English?

Reviewer #1: Yes

Reviewer #2: Yes

5. Review Comments to the Author

Reviewer #1: The manuscript in written in a clear and concise manner. The content is interesting and practical.

I think the topic is innovative and has implications for the medical field in general.

To improve an international audience understanding of content, add definitions for primary, secondary and tertiary facilities.

Also, replace to “our knowledge: with “based on the literature review for this project, this is only…”

Overall, this is a strong manuscript with the potential for a wide audience of readers.

Reviewer #2: This is a cross sectional study of healthcare professional views and experiences about health promotion and disease prevention in one municipality in South Africa. The authors find that although HCP value HP and DP, low levels of training in HP and DP were common and associated with lower levels of coordinated care and inclusion of HP and DP principles in medical care. A few comments/questions for the authors:

1. Why was this municipality chosen? How many HCP work in this municipality overall?

2. How were the 500 HCP chosen to participate? Convenience sample? What was the response rate?

3. Typically any results are reported as x/y (z%)

4. How did the questions decide on the survey questions to include? Was this based on any prior validated survey? Was the survey pilot tested for face or content validity?

5. P11 "The risk of coodinated HP training..." wording is awkward; would rewrite.

6. It would be helpful for Table 1 to list the demographics of the

7. The tables could be laid out more clearly and list the overall % of each group who agreed with the various statements rather than limited to Coefficients.

6. PLOS authors have the option to publish the peer review history of their article (what does this mean?). If published, this will include your full peer review and any attached files.

Reviewer #1: **Yes: **Michelle Lee D'Abundo

Reviewer #2: No

---

## [Author Response · Author response to Decision Letter 0]

15 Oct 2021

Reviewer 1 comments and responses:

To improve an international audience understanding of content, add definitions for primary, secondary, and tertiary facilities.

Response:

A primary health care facility refers to facilities where ambulatory or first-contact personal health care services are provided (26) and in this context includes the Primary Healthcare Centres (PHC), community clinics, Comprehensive Health Centers (CHC), and Midwife Obstetrics units (MOU). A secondary-level hospital refers to a healthcare facility that is highly differentiated by function with five to ten clinical specialties (27). This level of facility includes the Regional Hospitals, Provincial Hospitals, and General Hospitals. The capacity ranges between 200 and 800 beds. Tertiary-level hospitals on the other hand are hospitals with highly specialized professionals and facilities such as cardiology, intensive care unit, and specialized imaging infrastructures among others (27). The clinical services of the tertiary healthcare facilities are highly differentiated by function and have capacity of between 300 and 1500 beds. This level of hospitals includes National Hospitals, Central Hospitals, and Academic, or teaching or University Hospitals. 

Replace to “our knowledge: with “based on the literature review for this project, this is only…”

Response:

As suggested by the reviewer, this change has been done 

Reviewer 2 Comments and responses:

Why was this municipality chosen? How many HCP work in this municipality overall?

Response:

The NMBM is the largest City in the Eastern Cape Province and a major economic player both for the province and South Africa as a whole (24). Like the rest of South Africa, the municipality’s healthcare system has all three levels of care comprising primary, secondary and tertiary in place. Despite availability of inexpensive measures such as dietary counselling, physical activity, and handwashing for HP and DP, HCPs do not have structured training and knowledge to effectively implement the measures (13). Furthermore, there is limited evidence on the HP and DP training and attitudes of HCPs in relation to HP and DP especially in resource constrained settings.

The municipality has an estimated 3500 healthcare professionals (27) serving a population of about 1.24 million.

How were the 500 HCP chosen to participate? Convenience sample? What was the response rate?

This quantitative cross-sectional study was carried out among 520 HCPs that were randomly sampled from 23 hospitals in the NMBM. Only 501 HCPs completed the survey. Of the 501 respondents, six HCPs did not include their professions and were excluded from analysis. A total of 495 HCPs were included in the final analysis. An initial sample size of 384 was reached using a confidence level of 95% and an error margin of 5%. This sample size was further adjusted by 25% to 480 to account for non-responding HCPs. 

The response rate for this study was 95% (n=495). 

Typically any results are reported as x/y (z%)

We agree with this suggestion and have included tables 1, 2 and 4 to address this. 

How did the questions decide on the survey questions to include? Was this based on any prior validated survey? Was the survey pilot tested for face or content validity?

A standardized questionnaire adapted from a previously validated study (23) was developed to determine the influence of HCPs’ training and attitude on their practice of HP and DP. The questionnaire was further subjected to a pilot test. 

"The risk of coordinated HP training for staff for medical doctors" wording is awkward; would rewrite.

Revised accordingly. “The risk of coordinated HP training for medical doctors” 

It would be helpful for Table 1 to list the demographics.

Thank you for pointing this out. Although we agree that this is an important consideration, we have summarized the socio-demographic characteristics in page 5 to keep our tables concise.

The tables could be laid out more clearly and list the overall % of each group who agreed with the various statements rather than limited to Coefficients.

We agree with this suggestion and have included tables 1,2 and 4 to address this.

---

## [Editor Report · Decision Letter 1]

29 Oct 2021

Training, Attitudes, and Practice (TAP) among healthcare professionals in the Nelson Mandela Bay municipality, South Africa: A Health Promotion and Disease prevention perspective

PONE-D-21-07503R1

Dear Mr. Melariri,

We’re pleased to inform you that your manuscript has been judged scientifically suitable for publication and will be formally accepted for publication once it meets all outstanding technical requirements.

Kind regards,

Michelle Lee D'Abundo, PhD

Guest Editor

PLOS ONE

Additional Editor Comments (optional):

The information provided describes the TAP process in detail from a health promotion perspective. The research methods and results are clearly described and provide application for practice.

I support the publication of this article.
---

## [Editor Report · Acceptance letter]

16 Nov 2021

PONE-D-21-07503R1 

Training, Attitudes, and Practice (TAP) among healthcare professionals in the Nelson Mandela Bay municipality, South Africa: </i>A Health Promotion and Disease prevention perspective</i> 

Dear Dr. Melariri:

I'm pleased to inform you that your manuscript has been deemed suitable for publication in PLOS ONE. Congratulations! Your manuscript is now with our production department. 

Kind regards, 

on behalf of

Dr. Michelle Lee D'Abundo 

Guest Editor

PLOS ONE